# Learning-Based Low-Rank Approximations

**Piotr Indyk**
CSAIL, MIT
indyk@mit.edu

**Ali Vakilian**[*]
University of Wisconsin - Madison
vakilian@wisc.edu

**Yang Yuan**[*]
Tsinghua University
yuanyang@tsinghua.edu.cn

## Abstract

We introduce a "learning-based" algorithm for the *low-rank decomposition* problem: given an $n \times d$ matrix $A$, and a parameter $k$, compute a rank-$k$ matrix $A'$ that minimizes the approximation loss $\|A - A'\|_F$. The algorithm uses a training set of input matrices in order to optimize its performance. Specifically, some of the most efficient approximate algorithms for computing low-rank approximations proceed by computing a projection $SA$, where $S$ is a sparse random $m \times n$ "sketching matrix", and then performing the singular value decomposition of $SA$. We show how to replace the random matrix $S$ with a "learned" matrix of the same sparsity to reduce the error.

Our experiments show that, for multiple types of data sets, a learned sketch matrix can substantially reduce the approximation loss compared to a random matrix $S$, sometimes by one order of magnitude. We also study mixed matrices where only some of the rows are trained and the remaining ones are random, and show that matrices still offer improved performance while retaining worst-case guarantees.

Finally, to understand the theoretical aspects of our approach, we study the special case of $m = 1$. In particular, we give an approximation algorithm for minimizing the empirical loss, with approximation factor depending on the stable rank of matrices in the training set. We also show generalization bounds for the sketch matrix learning problem.

## 1 Introduction

The success of modern machine learning made it applicable to problems that lie outside of the scope of "classic AI". In particular, there has been a growing interest in using machine learning to improve the performance of "standard" algorithms, by fine-tuning their behavior to adapt to the properties of the input distribution, see e.g., [Wang et al., 2016, Khalil et al., 2017, Kraska et al., 2018, Balcan et al., 2018, Lykouris and Vassilvitskii, 2018, Purohit et al., 2018, Gollapudi and Panigrahi, 2019, Mitzenmacher, 2018, Mousavi et al., 2015, Baldassarre et al., 2016, Bora et al., 2017, Metzler et al., 2017, Hand and Voroninski, 2018, Khani et al., 2019, Hsu et al., 2019]. This "learning-based" approach to algorithm design has attracted a considerable attention over the last few years, due to its potential to significantly improve the efficiency of some of the most widely used algorithmic tasks. Many applications involve processing streams of data (video, data logs, customer activity etc) by executing the same algorithm on an hourly, daily or weekly basis. These data sets are typically not "random" or "worst-case"; instead, they come from some distribution which does not change rapidly from execution to execution. This makes it possible to design better algorithms tailored to the specific data distribution, trained on past instances of the problem.

The method has been particularly successful in the context of *compressed sensing*. In the latter framework, the goal is to recover an approximation to an $n$-dimensional vector $x$, given its "linear measurement" of the form $Sx$, where $S$ is an $m \times n$ matrix. Theoretical results [Donoho, 2006,

---

[*]This work was mostly done when the second and third authors were at MIT.

Candès et al., 2006] show that, if the matrix $S$ is selected *at random*, it is possible to recover the $k$ largest coefficients of $x$ with high probability using a matrix $S$ with $m = O(k \log n)$ rows. This guarantee is general and applies to arbitrary vectors $x$. However, if vectors $x$ are selected from some natural distribution (e.g., they represent images), recent works [Mousavi et al., 2015, Baldassarre et al., 2016, Metzler et al., 2017] show that one can use samples from that distribution to compute matrices $S$ that improve over a completely random matrix in terms of the recovery error.

Compressed sensing is an example of a broader class of problems which can be solved using random projections. Another well-studied problem of this type is *low-rank decomposition*: given an $n \times d$ matrix $A$, and a parameter $k$, compute a rank-$k$ matrix

$$[A]_k = \text{argmin}_{A': \, \text{rank}(A') \leq k} \|A - A'\|_F.$$

Low-rank approximation is one of the most widely used tools in massive data analysis, machine learning and statistics, and has been a subject of many algorithmic studies. In particular, multiple algorithms developed over the last decade use the "sketching" approach, see e.g., [Sarlos, 2006, Woolfe et al., 2008, Halko et al., 2011, Clarkson and Woodruff, 2009, 2017, Nelson and Nguyên, 2013, Meng and Mahoney, 2013, Boutsidis and Gittens, 2013, Cohen et al., 2015]. Its idea is to use efficiently computable random projections (a.k.a., "sketches") to reduce the problem size before performing low-rank decomposition, which makes the computation more space and time efficient. For example, [Sarlos, 2006, Clarkson and Woodruff, 2009] show that if $S$ is a random matrix of size $m \times n$ chosen from an appropriate distribution[2], for $m$ depending on $\epsilon$, then one can recover a rank-$k$ matrix $A'$ such that

$$\|A - A'\|_F \leq (1 + \epsilon)\|A - [A]_k\|_F$$

by performing an SVD on $SA \in \mathbb{R}^{m \times d}$ followed by some post-processing. Typically the sketch length $m$ is small, so the matrix $SA$ can be stored using little space (in the context of streaming algorithms) or efficiently communicated (in the context of distributed algorithms). Furthermore, the SVD of $SA$ can be computed efficiently, especially after another round of sketching, reducing the overall computation time. See the survey [Woodruff, 2014] for an overview of these developments.

In light of the aforementioned work on learning-based compressive sensing, it is natural to ask whether similar improvements in performance could be obtained for other sketch-based algorithms, notably for low-rank decompositions. In particular, reducing the sketch length $m$ while preserving its accuracy would make sketch-based algorithms more efficient. Alternatively, one could make sketches more accurate for the same values of $m$. This is the problem we address in this paper.

**Our Results.** Our main finding is that learned sketch matrices can indeed yield (much) more accurate low-rank decompositions than purely random matrices. We focus our study on a streaming algorithm for low-rank decomposition due to [Sarlos, 2006, Clarkson and Woodruff, 2009], described in more detail in Section 2. Specifically, suppose we have a training set of matrices $\text{Tr} = \{A_1, \ldots, A_N\}$ sampled from some distribution D. Based on this training set, we compute a matrix $S^*$ that (locally) minimizes the empirical loss

$$\sum_i \|A_i - \text{SCW}(S^*, A_i)\|_F \tag{1}$$

where $\text{SCW}(S^*, A_i)$ denotes the output of the aforementioned Sarlos-Clarkson-Woodruff streaming low-rank decomposition algorithm on matrix $A_i$ using the sketch matrix $S^*$. Once the sketch matrix $S^*$ is computed, it can be used instead of a random sketch matrix in all future executions of the SCW algorithm.

We demonstrate empirically that, for multiple types of data sets, an optimized sketch matrix $S^*$ can substantially reduce the approximation loss compared to a random matrix $S$, sometimes by one order of magnitude (see Figure 2 or 3). Equivalently, the optimized sketch matrix can achieve the same approximation loss for lower values of $m$ which results in sketching matrices with lower space usage. Note that since we augment a streaming algorithm, our main focus is on improving its *space* usage

(which in the distributed setting translates into the amount of communication). The latter is $O(md)$, the size of $SA$.

A possible disadvantage of learned sketch matrices is that an algorithm that uses them no longer offers *worst-case* guarantees. As a result, if such an algorithm is applied to an input matrix that does not conform to the training distribution, the results might be worse than if random matrices were used. To alleviate this issue, we also study *mixed* sketch matrices, where (say) half of the rows are trained and the other half are random. We observe that if such matrices are used in conjunction with the SCW algorithm, its results are no worse than if only the random part of the matrix was used (Theorem 1 in Section 4)[3]. Thus, the resulting algorithm inherits the worst-case performance guarantees of the random part of the sketching matrix. At the same time, we show that mixed matrices still substantially reduce the approximation loss compared to random ones, in some cases nearly matching the performance of "pure" learned matrices with the same number of rows. Thus, mixed random matrices offer "the best of both worlds": improved performance for matrices from the training distribution, and worst-case guarantees otherwise.

Finally, in order to understand the theoretical aspects of our approach further, we study the special case of $m = 1$. This corresponds to the case where the sketch matrix $S$ is just a single vector. Our results are two-fold:

- We give an approximation algorithm for minimizing the empirical loss as in Equation 1, with an approximation factor depending on the stable rank of matrices in the training set. See Appendix B.
- Under certain assumptions about the robustness of the loss minimizer, we show generalization bounds for the solution computed over the training set. See Appendix C.

The theoretical results on the case of $m = 1$ are deferred to the full version of this paper.

## 1.1 Related work

As outlined in the introduction, over the last few years there has been multiple papers exploring the use of machine learning methods to improve the performance of "standard" algorithms. Among those, the closest to the topic of our paper are the works on learning-based compressive sensing, such as [Mousavi et al., 2015, Baldassarre et al., 2016, Bora et al., 2017, Metzler et al., 2017], and on learning-based streaming algorithms [Hsu et al., 2019]. Since neither of these two lines of research addresses computing matrix spectra, the technical development therein was quite different from ours.

In this paper we focus on learning-based optimization of low-rank approximation algorithms that use *linear sketches*, i.e., map the input matrix $A$ into $SA$ and perform computation on the latter. There are other sketching algorithms for low-rank approximation that involve *non-linear* sketches [Liberty, 2013, Ghashami and Phillips, 2014, Ghashami et al., 2016]. The benefit of linear sketches is that they are easy to update under linear changes to the matrix $A$, and (in the context of our work) that they are easy to differentiate, making it possible to compute the gradient of the loss function as in Equation 1. We do not know whether it is possible to use our learning-based approach for non-linear sketches, but we believe this is an interesting direction for future research.

## 2 Preliminaries

**Notation.** Consider a distribution D on matrices $A \in \mathbb{R}^{n \times d}$. We define the training set as $\{A_1, \cdots, A_N\}$ sampled from D. For matrix $A$, its *singular value decomposition (SVD)* can be written as $A = U \Sigma V^\top$ such that both $U \in \mathbb{R}^{n \times n}$ and $V \in \mathbb{R}^{d \times n}$ have *orthonormal columns* and $\Sigma = \text{diag}\{\lambda_1, \cdots, \lambda_d\}$ is a diagonal matrix with nonnegative entries. Moreover, if $\text{rank}(A) = r$, then the first $r$ columns of $U$ are an orthonormal basis for the *column space* of $A$ (we denote it as $\text{colsp}(A)$), the first $r$ columns of $V$ are an orthonormal basis for the *row space* of $A$ (we denote it as $\text{rowsp}(A)$)[4] and $\lambda_i = 0$ for $i > r$. In many applications it is quicker and more economical to compute the *compact SVD* which only contains the rows and columns corresponding to the non-zero singular values of $\Sigma$: $A = U^c \Sigma^c (V^c)^\top$ where $U^c \in \mathbb{R}^{n \times r}, \Sigma^c \in \mathbb{R}^{r \times r}$ and $V^c \in \mathbb{R}^{d \times r}$.

**How sketching works.** We start by describing the SCW algorithm for low-rank matrix approximation, see Algorithm 1. The algorithm computes the singular value decomposition of $SA = U\Sigma V^\top$, and compute the best rank-$k$ approximation of $AV$. Finally it outputs $[AV]_k V^\top$ as a rank-$k$ approximation of $A$. We emphasize that Sarlos and Clarkson-Woodruff proposed Algorithm 1 with random sketching matrices $S$. In this paper, we follow the same framework but use learned (or partially learned) matrices.

---

**Algorithm 1** Rank-$k$ approximation of a matrix $A$ using a sketch matrix $S$ (refer to Section 4.1.1 of [Clarkson and Woodruff, 2009])

---

1: **Input:** $A \in \mathbb{R}^{n\times d}, S \in \mathbb{R}^{m\times n}$
2: $U, \Sigma, V^\top \leftarrow \text{COMPACTSVD}(SA)$    $\triangleright$    $\{r = \text{rank}(SA), U \in \mathbb{R}^{m\times r}, V \in \mathbb{R}^{d\times r}\}$
3: **Return:** $[AV]_k V^\top$

---

Note that if $m$ is much smaller than $d$ and $n$, the space bound of this algorithm is significantly better than when computing a rank-$k$ approximation for $A$ in the naïve way. Thus, minimizing $m$ automatically reduces the space usage of the algorithm.

**Sketching matrix.** We use matrix $S$ that is sparse[5] Specifically, each column of $S$ has exactly one non-zero entry, which is either $+1$ or $-1$. This means that the fraction of non-zero entries in $S$ is $1/m$. Therefore, one can use a vector to represent $S$, which is very memory efficient. It is worth noting, however, after multiplying the sketching matrix $S$ with other matrices, the resulting matrix (e.g., $SA$) is in general not sparse.

# 3 Training Algorithm

In this section, we describe our learning-based algorithm for computing a data dependent sketch $S$. The main idea is to use backpropagation algorithm to compute the stochastic gradient of $S$ with respect to the rank-$k$ approximation loss in Equation 1, where the initial value of $S$ is the same random sparse matrix used in SCW. Once we have the stochastic gradient, we can run stochastic gradient descent (SGD) algorithm to optimize $S$, in order to improve the loss. Our algorithm maintains the sparse structure of $S$, and only optimizes the values of the $n$ non-zero entries (initially $+1$ or $-1$).

---

**Algorithm 2** Differentiable SVD implementation

---

1: **Input:** $A_1 \in \mathbb{R}^{m\times d}(m < d)$
2: $U, \Sigma, V \leftarrow \{\}, \{\}, \{\}$
3: **for** $i \leftarrow 1 \dots m$ **do**
4:     $v_1 \leftarrow$ random initialization in $\mathbb{R}^d$
5:     **for** $t \leftarrow 1 \dots T$ **do**
6:        $v_{t+1} \leftarrow \frac{A_i^\top A_i v_t}{\|A_i^\top A_i v_t\|_2}$    $\triangleright$    {power method}
7:     **end for**
8:     $V[i] \leftarrow v_{T+1}$
9:     $\Sigma[i] \leftarrow \|A_i V[i]\|_2$
10:     $U[i] \leftarrow \frac{A_i V[i]}{\Sigma[i]}$
11:     $A_{i+1} \leftarrow A_i - \Sigma[i]U[i]V[i]^\top$
12: **end for**
13: **Return:** $U, \Sigma, V$

---

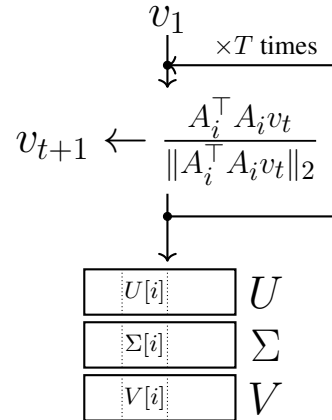

Figure 1: $i$-th iteration of power method

However, the standard SVD implementation (step 2 in Algorithm 1 ) is not differentiable, which means we cannot get the gradient in the straightforward way. To make SVD implementation differentiable, we use the fact that the SVD procedure can be represented as $m$ individual top singular value decompositions (see e.g. [Allen-Zhu and Li, 2016]), and that every top singular value decomposition

can be computed using the power method. See Figure 1 and Algorithm 2. We store the results of the $i$-th iteration into the $i$-th entry of the list $U, \Sigma, V$, and finally concatenate all entries together to get the matrix (or matrix diagonal) format of $U, \Sigma, V$. This allows gradients to flow easily.

Due to the extremely long computational chain, it is infeasible to write down the explicit form of loss function or the gradients. However, just like how modern deep neural networks compute their gradients, we used the autograd feature in PyTorch to *numerically* compute the gradient with respect to the sketching matrix $S$.

We emphasize again that our method is only optimizing $S$ for the training phase. After $S$ is fully trained, we still call Algorithm 1 for low rank approximation, which has exactly the same running time as the SCW algorithm, but with better performance (i.e., the quality of the returned rank-$k$ matrix). We remark that the *time* complexity of SCW algorithm is $O(nmd)$ assuming $k \leq m \leq \min(n, d)$.

## 4 Worst Case Bound

In this section, we show that concatenating two sketching matrices $S_1$ and $S_2$ (of size respectively $m_1 \times n$ and $m_2 \times n$) into a single matrix $S_*$ (of size $(m_1 + m_2) \times n$) will not increase the approximation loss of the final rank-$k$ solution computed by Algorithm 1 compared to the case in which only one of $S_1$ or $S_2$ are used as the sketching matrix. In the rest of this section, the sketching matrix $S_*$ denotes the concatenation of $S_1$ and $S_2$ as follows:

$$ S_{*((m_1+m_2)\times n)} = \left[ \begin{array}{c} S_{1(m_1 \times n)} \\ \rule{3cm}{2pt} \\ S_{2(m_2 \times n)} \end{array} \right] $$

Formally, we prove the following theorem on the worst case performance of *mixed matrices*.

**Theorem 1.** *Let $U_* \Sigma_* V_*^\top$ and $U_1 \Sigma_1 V_1^\top$ respectively denote the SVD of $S_* A$ and $S_1 A$. Then,*

$$ ||[AV_*]_k V_*^\top - A||_F \leq ||[AV_1]_k V_1^\top - A||_F. $$

In particular, the above theorem implies that the output of Algorithm 1 with the sketching matrix $S_*$ is a better rank-$k$ approximation to $A$ compared to the output of the algorithm with $S_1$. In the rest of this section we prove Theorem 1.

Before proving the main theorem, we state the following helpful lemma.

**Lemma 1** (Lemma 4.3 in [Clarkson and Woodruff, 2009]). *Suppose that $V$ is a matrix with orthonormal columns. Then, a best rank-k approximation to $A$ in the $\mathrm{colsp}(V)$ is given by $[AV]_k V^\top$.*

Since the above statement is a transposed version of the lemma from [Clarkson and Woodruff, 2009], we include the proof in the appendix for completeness.

*Proof of Theorem 1.* First, we show that $\mathrm{colsp}(V_1) \subseteq \mathrm{colsp}(V_*)$. By the properties of the (compact) SVD, $\mathrm{colsp}(V_1) = \mathrm{rowsp}(S_1 A)$ and $\mathrm{colsp}(V_*) = \mathrm{rowsp}(S_* A)$. Since, $S_*$ has all rows of $S_1$, then

$$ \mathrm{colsp}(V_1) \subseteq \mathrm{colsp}(V_*). \tag{2} $$

By Lemma 1,

$$ ||A - [AV_*]_k V_*^\top||_F = \min_{\substack{\mathrm{rowsp}(X) \subseteq \mathrm{colsp}(V_*); \\ \mathrm{rank}(X) \leq k}} ||X - A||_F $$

$$ ||A - [AV_1]_k V_1^\top||_F = \min_{\substack{\mathrm{rowsp}(X) \subseteq \mathrm{colsp}(V_1); \\ \mathrm{rank}(X) \leq k}} ||X - A||_F $$

Finally, together with (2),

$$ ||A - [AV_*]_k V_*^\top||_F = \min_{\substack{\mathrm{rowsp}(X) \subseteq \mathrm{colsp}(V_*); \\ \mathrm{rank}(X) \leq k}} ||X - A||_F $$

$$ \leq \min_{\substack{\mathrm{rowsp}(X) \subseteq \mathrm{colsp}(V_1); \\ \mathrm{rank}(X) \leq k}} ||X - A||_F = ||A - [AV_1]_k V_1^\top||_F. $$

which completes the proof. □

Finally, we note that the property of Theorem 1 is not universal, i.e., it does not hold for all sketching algorithms for low-rank decomposition. For example, an alternative algorithm proposed in [Cohen et al., 2015] proceeds by letting $Z$ to be the top $k$ singular vectors of $SA$ (i.e., $Z = V$ where $[SA]_k = U\Sigma V^T$) and then reports $AZZ^\top$. It is not difficult to see that, by adding extra rows to the sketching matrix $S$ (which may change all top $k$ singular vectors compared to the ones of $SA$), one can skew the output of the algorithm so that it is far from the optimal.

## 5 Experimental Results

The main question considered in this paper is whether, for natural matrix datasets, optimizing the sketch matrix $S$ can improve the performance of the sketching algorithm for the low-rank decomposition problem. To answer this question, we implemented and compared the following methods for computing $S \in \mathbb{R}^{m \times n}$.

- **Sparse Random**. Sketching matrices are generated at random as in [Clarkson and Woodruff, 2017]. Specifically, we select a random hash function $h : [n] \rightarrow [m]$, and for all $i = 1 \ldots n$, $S_{h[i],i}$ is selected to be either $+1$ or $-1$ with equal probability. All other entries in $S$ are set to $0$. Therefore, $S$ has exactly $n$ non-zero entries.
- **Dense Random**. All the $nm$ entries in the sketching matrices are sampled from Gaussian distribution (we include this method for comparison).
- **Learned**. Using the sparse random matrix as the initialization, we run Algorithm 2 to optimize the sketching matrix using the training set, and return the optimized matrix.
- **Mixed (J)**. We first generate two sparse random matrices $S_1, S_2 \in \mathbb{R}^{\frac{m}{2} \times n}$ (assuming $m$ is even), and define $S$ to be their combination. We then run Algorithm 2 to optimize $S$ using the training set, but only $S_1$ will be updated, while $S_2$ is fixed. Therefore, $S$ is a mixture of learned matrix and random matrix, and the first matrix is trained *jointly* with the second one.
- **Mixed (S)**. We first compute a learned matrix $S_1 \in \mathbb{R}^{\frac{m}{2} \times n}$ using the training set, and then append another sparse random matrix $S_2$ to get $S \in \mathbb{R}^{m \times n}$. Therefore, $S$ is a mixture of learned matrix and random matrix, but the learned matrix is trained *separately*.

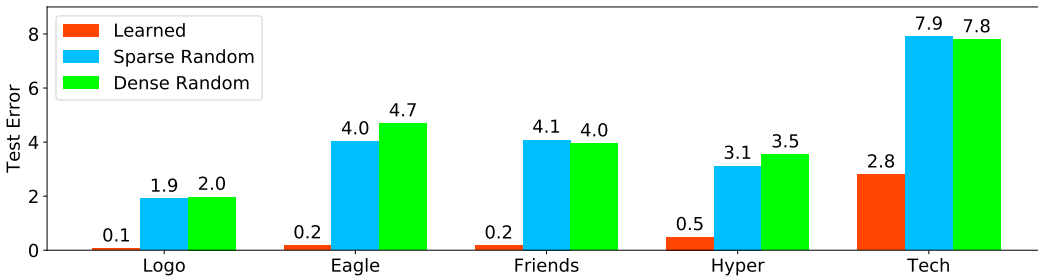

Figure 2: Test error by datasets and sketching matrices For $k = 10, m = 20$

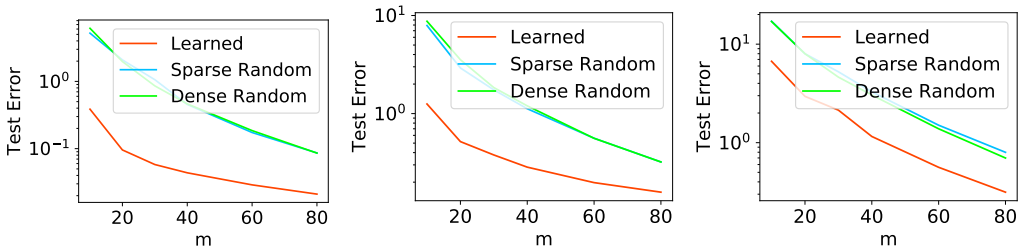

Figure 3: Test error for Logo (left), Hyper (middle) and Tech (right) when $k = 10$.

**Datasets.**    We used a variety of datasets to test the performance of our methods:

| Table 1: Test error in various settings | | | | | |
| --- | --- | --- | --- | --- | --- |
| $k, m$, Sketch | Logo | Eagle | Friends | Hyper | Tech |
| 10, 10, Learned | **0.39** | **0.31** | **1.03** | **1.25** | **6.70** |
| 10, 10, Random | 5.22 | 6.33 | 11.56 | 7.90 | 17.08 |
| 10, 20, Learned | **0.10** | **0.18** | **0.22** | **0.52** | **2.95** |
| 10, 20, Random | 2.09 | 4.31 | 4.11 | 2.92 | 7.99 |
| 20, 20, Learned | **0.61** | **0.66** | **1.41** | **1.68** | **7.79** |
| 20, 20, Random | 4.18 | 5.79 | 9.10 | 5.71 | 14.55 |
| 20, 40, Learned | **0.18** | **0.41** | **0.42** | **0.72** | **3.09** |
| 20, 40, Random | 1.19 | 3.50 | 2.44 | 2.23 | 6.20 |
| 30, 30, Learned | **0.72** | **1.06** | **1.78** | **1.90** | **7.14** |
| 30, 30, Random | 3.11 | 6.03 | 6.27 | 5.23 | 12.82 |
| 30, 60, Learned | **0.21** | **0.61** | **0.42** | **0.84** | **2.78** |
| 30, 60, Random | 0.82 | 3.28 | 1.79 | 1.88 | 4.84 |

| Table 2: Comparison with mixed sketches | | | |
| --- | --- | --- | --- |
| $k, m$, Sketch | Logo | Hyper | Tech |
| 10, 10, Learned | **0.39** | **1.25** | **6.70** |
| 10, 10, Random | 5.22 | 7.90 | 17.08 |
| 10, 20, Learned | **0.10** | **0.52** | **2.95** |
| 10, 20, Mixed (J) | 0.20 | 0.78 | 3.73 |
| 10, 20, Mixed (S) | 0.24 | 0.87 | 3.69 |
| 10, 20, Random | 2.09 | 2.92 | 7.99 |
| 10, 40, Learned | **0.04** | **0.28** | **1.16** |
| 10, 40, Mixed (J) | 0.05 | 0.34 | 1.31 |
| 10, 40, Mixed (S) | 0.05 | 0.34 | 1.20 |
| 10, 40, Random | 0.45 | 1.12 | 3.28 |
| 10, 80, Learned | **0.02** | **0.16** | **0.31** |
| 10, 80, Random | 0.09 | 0.32 | 0.80 |

- **Videos[6]: Logo, Friends, Eagle.** We downloaded three high resolution videos from Youtube, including logo video, Friends TV show, and eagle nest cam. From each video, we collect $500$ frames of size $1920 \times 1080 \times 3$ pixels, and use $400$ ($100$) matrices as the training (test) set. For each frame, we resize it as a $5760 \times 1080$ matrix.

- **Hyper.** We use matrices from HS-SOD, a dataset for hyperspectral images from natural scenes [Imamoglu et al., 2018]. Each matrix has $1024 \times 768$ pixels, and we use $400$ ($100$) matrices as the training (test) set.

- **Tech.** We use matrices from TechTC-300, a dataset for text categorization [Davidov et al., 2004]. Each matrix has $835, 422$ rows, but on average only $25, 389$ of the rows contain non-zero entries. On average each matrix has $195$ columns. We use $200$ ($95$) matrices as the training (test) set.

**Evaluation metric.** To evaluate the quality of a sketching matrix $S$, it suffices to evaluate the output of Algorithm 1 using the sketching matrix $S$ on different input matrices $A$. We first define the optimal approximation loss for test set Te as follows: $\text{App}^*_{\text{Te}} \triangleq \mathbf{E}_{A \sim \text{Te}} \|A - [A]_k\|_F$.

Note that $\text{App}^*_{\text{Te}}$ does not depend on $S$, and in general it is not achievable by any sketch $S$ with $m < d$, because of information loss. Based on the definition of the optimal approximation loss, we define the error of the sketch $S$ for Te as $\text{Err}(\text{Te}, S) \triangleq \mathbf{E}_{A \sim \text{Te}} \|A - \text{SCW}(S, A)\|_F - \text{App}^*_{\text{Te}}$.

In our datasets, some of the matrices have much larger singular values than the others. To avoid imbalance in the dataset, we normalize the matrices so that their top singular values are all equal.

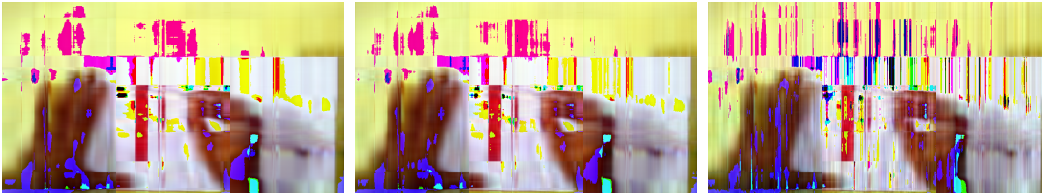

Figure 4: Low rank approximation results for Logo video frame: the best rank-10 approximation (left), and rank-10 approximations reported by Algorithm 1 using a sparse learned sketching matrix (middle) and a sparse random sketching matrix (right).

## 5.1 Average test error

We first test all methods on different datasets, with various combination of $k, m$. See Figure 2 for the results when $k = 10, m = 20$. As we can see, for video datasets, learned sketching matrices can get $20 \times$ better test error than the sparse random or dense random sketching matrices. For other

datasets, learned sketching matrices are still more than $2\times$ better. In this experiment, we have run each configuration 5 times, and computed the standard error of each test error[7]. For Logo, Eagle, Friends, Hyper and Tech, the standard errors of *learned*, *sparse random* and *dense random* sketching matrices are respectively, $(1.5, 8.4, 35.3, 124, 41) \times 10^{-6}$, $(3.1, 5.3, 7.0, 2.9, 4.5) \times 10^{-2}$ and $(3.5, 18.1, 4.6, 10.7, 3.3) \times 10^{-2}$. It is clear that the standard error of the learned sketching matrix is a few order of magnitudes smaller than the random sketching matrices, which shows another benefit of learning sketching matrices.

Similar improvement of the learned sketching matrices over the random sketching matrices can be observed when $k = 10, m = 10, 20, 30, 40, \cdots, 80$, see Figure 3. We also include the test error results in Table 1 for the case when $k = 20, 30$. Finally, in Figure 4, we visualize an example output of the algorithm for the case $k = 10, m = 20$ for the Logo dataset.

## 5.2 Comparing Random, Learned and Mixed

In Table 2, we investigate the performance of the mixed sketching matrices by comparing them with random and learned sketching matrices. In all scenarios, the mixed sketching matrices yield much better results than the random sketching matrices, and sometimes the results are comparable to those of learned sketching matrices. This means, in most cases it suffices to train half of the sketching matrix to obtain good empirical results, and at the same time, by our Theorem 1, we can use the remaining random half of the sketching matrix to obtain worst-case guarantees.

Moreover, if we do not fix the number of learned rows to be half, the test error increases as the number of learned rows decreases. In Figure 5, we plot the test error for the setting with $m = 20, k = 10$ using 100 Logo matrices, running for 3000 iterations.

## 5.3 Mixing Training Sets

In our previous experiments, we constructed a different learned sketching matrix $S$ for each data set. However, one can use a *single* random sketching matrix for all three data sets simultaneously. Next, we study the performance of a *single* learned sketching matrix for all three data sets. In Table 3, we constructed a single learned sketching matrix $S$ with $m = k = 10$ on a training set containing 300 matrices from Logo, Eagle and Friends (each has 100 matrices). Then, we tested $S$ on Logo matrices and compared its performance to the performance of a learned sketching matrix $S_L$ trained on Logo dataset (i.e., using 100 Logo matrices only), as well as to the performance of a random sketching $S_R$. The performance of the sketching matrix $S$ with a mixed training set from all three datasets is close to the performance of the sketching matrix $S_L$ with training set only from Logo dataset, and is much better than the performance of the random sketching matrix $S_R$.

## 5.4 Running Time

The runtimes of the algorithm with a random sketching matrix and our learned sketching matrix are the same, and are much less than the runtime of the "standard" SVD method (implemented in Pytorch). In Table 4, we present the runtimes of the algorithm with different types of sketching matrices (i.e., *learned* and *random*) on Logo matrices with $m = k = 10$, as well as the training time of the learned case. Notice that training only needs to be done *once*, and can be done offline.

# 6 Conclusions

In this paper we introduced a learning-based approach to sketching algorithms for computing low-rank decompositions. Such algorithms proceed by computing a projection $SA$, where $A$ is the input matrix and $S$ is a random "sketching" matrix. We showed how to train $S$ using example matrices $A$ in order to improve the performance of the overall algorithm. Our experiments show that for several different types of datasets, a learned sketch can significantly reduce the approximation loss compared to a random matrix. Further, we showed that if we mix a random matrix and a learned matrix (by concatenation), the result still offers an improved performance while inheriting worst case guarantees of the random sketch component.

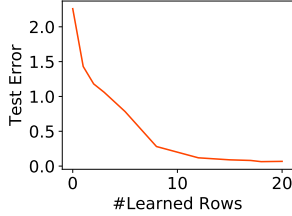

Figure 5: Test errors of mixed sketching matrices with different number of "learned" rows.

Table 3: Evaluation of the sketching matrix trained on different sets

|  | Logo+Eagle+Friends | Logo only | Random |
|---|---|---|---|
| Test Error | 0.67 | 0.27 | 5.19 |

Table 4: Runtimes of the algorithm with different sketching matrices

| SVD | Random | Learned-Inference | Learned-Training |
|---|---|---|---|
| 2.2s | 0.03s | 0.03s | 9481.25s |

# Acknowledgment

This research was supported by NSF TRIPODS award #1740751 and Simons Investigator Award. The authors would like to thank the anonymous reviewers for their insightful comments and suggestions.

## Footnotes

[2]Initial algorithms used matrices with independent sub-gaussian entries or randomized Fourier/Hadamard matrices [Sarlos, 2006, Woolfe et al., 2008, Halko et al., 2011]. Starting from the seminal work of [Clarkson and Woodruff, 2017], researchers began to explore sparse binary matrices, see e.g., [Nelson and Nguyên, 2013, Meng and Mahoney, 2013]. In this paper we mostly focus on the latter distribution.

[3]We note that this property is non-trivial, in the sense that it does not automatically hold for *all* sketching algorithms. See Section 4 for further discussion.

[4]The remaining columns of $U$ and $V$ respectively are orthonormal bases for the nullspace of $A$ and $A^\top$.

[5]The original papers [Sarlos, 2006, Clarkson and Woodruff, 2009] used dense matrices, but the work of [Clarkson and Woodruff, 2017] showed that sparse matrices work as well. We use sparse matrices since they are more efficient to train and to operate on.

[6]They can be downloaded from `http://youtu.be/L5HQoFIaT4I`, `http://youtu.be/xmLZsEfXEgE` and `http://youtu.be/ufnf_q_30fg`

[7]They were very small, so we did not plot in the figures

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
