[Supplementary Material]

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

# A   The case of $m = 1$

In this section, we denote the SVD of $A$ as $U^A \Sigma^A (V^A)^\top$ such that both $U^A$ and $V^A$ have *or-thonormal columns* and $\Sigma^A = \mathrm{diag}\{\lambda_1^A, \cdots, \lambda_d^A\}$ is a diagonal matrix with nonnegative entries. For simplicity, we assume that for all $A \sim \mathsf{D}$, $1 = \lambda_1 \geq \cdots \geq \lambda_d$. We use $U_i^A$ to denote the $i$-th column of $U^A$, and similarly for $V_i^A$. Denote $\Sigma^A = \mathrm{diag}\{\lambda_1^A, \cdots, \lambda_d^A\}$.

We want to find $[A]_k$, the rank-$k$ approximation of $A$. In general, it is hard to obtain a closed form expression of the output of Algorithm 1. However, for $m = 1$, such expressions can be calculated. Indeed, if $m = 1$, the sketching matrix becomes a vector $s \in \mathbb{R}^{1 \times n}$. Therefore $[AV]_k$ has rank at most one, so it suffices to set $k = 1$. Consider a matrix $A \sim \mathsf{D}$ as the input to Algorithm 1. By calculation, $SA = \sum_i \lambda_i^A \langle s, U_i^A \rangle (V_i^A)^\top$, which is a vector. For example, if $S = U_1^A$, we obtain $\lambda_1^A (V_1^A)^\top$. Note that in this section to emphasize that $m = 1$ (i.e., $S$ is a vector), we refer to $S$ as $s$.

Since $SA$ is a vector, applying SVD on it is equivalent to performing normalization. Therefore,

$$V = \frac{\sum_{i=1}^d \lambda_i^A \langle s, U_i^A \rangle (V_i^A)^\top}{\sqrt{\sum_{i=1}^d (\lambda_i^A)^2 \langle s, U_i^A \rangle^2}}$$

Ideally, we hope that $V$ is as close to $V_1^A$ as possible, because that means $[AV]_1 V^\top$ is close to $\lambda_1^A U_1^A (V_1^A)^\top$, which captures the top singular component of $A$, i.e., the optimal solution. More formally,

$$AV = \frac{\sum_{i=1}^d (\lambda_i^A)^2 \langle s, U_i^A \rangle U_i^A}{\sqrt{\sum_{i=1}^d (\lambda_i^A)^2 \langle s, U_i^A \rangle^2}}$$

We want to maximize its norm, which is:

$$\frac{\sum_{i=1}^d (\lambda_i^A)^4 \langle s, U_i^A \rangle^2}{\sum_{i=1}^d (\lambda_i^A)^2 \langle s, U_i^A \rangle^2} \tag{3}$$

We note that one can simplify (3) by considering only the contribution from the top left singular vector $U_1^A$, which corresponds to the maximization of the following expression:

$$\frac{\langle s, U_1^A \rangle^2}{\sum_{i=1}^d (\lambda_i^A)^2 \langle s, U_i^A \rangle^2} \tag{4}$$

# B   Optimization Bounds

Motivated by the empirical success of sketch optimization, we investigate the complexity of optimizing the loss function. We focus on the simple case where $m = 1$ and therefore $S$ is just a (dense) vector. Our main observation is that a vector $s$ picked *uniformly at random* from the $d$-dimensional unit sphere achieves an approximately optimal solution, with the approximation factor depending on the maximum *stable rank* of matrices $A_1, \cdots, A_N$. This algorithm is not particularly useful for our purpose, as our goal is to *improve* over the random choice of the sketching matrix $S$. Nevertheless, it demonstrates that an algorithm with a non-trivial approximation factor exists.

**Definition 1** (stable rank $(r')$). *For a matrix $A$, the stable rank of $A$ is defined as the squared ratio of Frobenius and operator norm of $A$. I.e.,*

$$r'(A) = \frac{\|A\|_F^2}{\|A\|_2^2} = \frac{\sum_i (\lambda_i^A)^2}{\max_i (\lambda_i^A)^2}.$$

Note that since we assume for all matrices $A \sim \mathsf{D}$, $1 = \lambda_1^A \geq \cdots \geq \lambda_d^A > 0$, for all these matrices $r'(A) = \sum_i (\lambda_i^A)^2$.

First, we consider the simplified objective function as in (4).

**Lemma 2.** *A random vector $s$ which is picked uniformly at random from the $d$-dimensional unit sphere, is an $O(r')$-approximation to the optimum value of the simplified objective function in Equation (4), where $r'$ is the maximum stable rank of matrices $A_1, \cdots, A_N$.*

*Proof.* We will show that

$$E\left[\frac{\langle s, U_1^A\rangle^2}{\sum_{i=1}^d (\lambda_i^A)^2 \langle s, U_i^A\rangle^2}\right] = \Omega(1/r'(A))$$

for all $A \sim \mathsf{D}$ where $s$ is a vector picked uniformly at random from $\mathbf{S}^{d-1}$. Since for all $A \sim \mathsf{D}$ we have $\frac{\langle s, U_1^A\rangle^2}{\sum_{i=1}^d (\lambda_i^A)^2 \langle s, U_i^A\rangle^2} \leq 1$, by the linearity of expectation we have that the vector $s$ achieves an $O(r')$-approximation to the maximum value of the objective function

$$\sum_{j=1}^N \frac{\langle s, U_1^{A_j}\rangle^2}{\sum_{i=1}^d (\lambda_i^{A_j})^2 \langle s, U_i^{A_j}\rangle^2}$$

.

First, recall that to sample $s$ uniformly at random from $\mathbf{S}^{d-1}$ we can generate $s$ as $\sum_{i=1}^d \alpha_i U_i^A / \sqrt{\sum_{i=1}^d \alpha_i^2}$ where for all $i \leq d$, $\alpha_i \sim \mathcal{N}(0,1)$. This helps us evaluate $\mathbf{E}\left[\frac{\langle s, U_1^A\rangle^2}{\sum_{i=1}^d (\lambda_i^A)^2 \langle s, U_i^A\rangle^2}\right]$ for an arbitrary matrix $A \sim \mathsf{D}$:

$$E = \mathbf{E}\left[\frac{\langle s, U_1^A\rangle^2}{\sum_{i=1}^d (\lambda_i^A)^2 \langle s, U_i^A\rangle^2}\right] = \mathbf{E}\left[\frac{(\alpha_1)^2}{\sum_{i=1}^d (\lambda_i^A)^2 \cdot (\alpha_i)^2}\right]$$

$$\geq \mathbf{E}\left[\frac{(\alpha_1)^2}{\sum_{i=1}^d (\lambda_i^A)^2 \cdot (\alpha_i)^2}|\Psi_1 \cap \Psi_2\right] \cdot \Pr(\Psi_1 \cap \Psi_2)$$

where the events $\Psi_1, \Psi_2$ are defined as:

$$\Psi_1 \triangleq \mathbf{1}\left[|\alpha_1| \geq \frac{1}{2}\right], \text{ and } \Psi_2 \triangleq \mathbf{1}\left[\sum_{i=2}^d (\lambda_i^A)^2 (\alpha_i)^2 \leq 2 \cdot r'(A)\right]$$

Since $\alpha_i$s are independent, we have

$$E \geq \mathbf{E}\left[\frac{(\alpha_1)^2}{(\alpha_1)^2 + 2 \cdot r'(A)}|\Psi_1 \cap \Psi_2\right] \cdot \Pr(\Psi_1) \cdot \Pr(\Psi_2) \geq \frac{1}{8 \cdot r'(A) + 1} \cdot \Pr(\Psi_1) \cdot \Pr(\Psi_2)$$

where we used that $\frac{(\alpha_1)^2}{(\alpha_1)^2 + 2 \cdot r'(A)}$ is increasing for $(\alpha_1)^2 \geq \frac{1}{4}$. It remains to prove that $\Pr(\Psi_1), \Pr(\Psi_2) = \Theta(1)$. We observe that, since $\alpha_i \sim \mathcal{N}(0,1)$, we have

$$\Pr(\Psi_1) = \Pr\left(|\alpha_1| \geq \frac{1}{2}\right) = \Theta(1)$$

Similarly, by Markov inequality, we have

$$\Pr(\Psi_2) = \Pr\left(\sum_{i=1}^d (\lambda_i^A)^2 (\alpha_i)^2 \leq 2r'(A)\right) \geq 1 - \Pr\left(\sum_{i=1}^d (\lambda_i^A)^2 (\alpha_i)^2 > 2r'(A)\right) \geq \frac{1}{2}$$

$\square$

Next, we prove that a random vector $s \in \mathbf{S}^{d-1}$ achieves an $O(r'(A))$-approximation to the optimum of the main objective function as in Equation (3).

**Lemma 3.** *A random vector $s$ which is picked uniformly at random from the d-dimensional unit sphere, is an $O(r')$-approximation to the optimum value of the objective function in Equation* (3)*, where $r'$ is the maximum stable rank of matrices $A_1, \cdots, A_N$.*

*Proof.* We assume that the vector $s$ is generated via the same process as in the proof of Lemma 2. It follows that

$$\mathbf{E}\left[\frac{\sum_{i=1}^d (\lambda_i^A)^4 \langle s, U_i^A\rangle^2}{\sum_{i=1}^d (\lambda_i^A)^2 \langle s, U_i^A\rangle^2}\right] \geq \mathbf{E}\left[\frac{(\alpha_1)^2}{\sum_{i=1}^d (\lambda_i^A)^2 \cdot (\alpha_i)^2}\right] = \Omega(1/r'(A))$$

$\square$

## C  Generalization Bounds

Define the loss function as

$$\mathsf{L}(s) \triangleq -\mathbf{E}_{A\sim\mathsf{D}}\left[\frac{\sum_{i=1}^{d}\left(\lambda_i^A\right)^4\langle s, U_i^A\rangle^2}{\sum_{i=1}^{d}\left(\lambda_i^A\right)^2\langle s, U_i^A\rangle^2}\right]$$

We want to find a vector $s \in \mathbf{S}^{d-1}$ to minimize $\mathsf{L}(s)$, where $\mathbf{S}^{d-1}$ is the $d$-dimensional unit sphere. Since $\mathsf{D}$ is unknown, we are optimizing the following empirical loss:

$$\hat{\mathsf{L}}_{\mathsf{Tr}}(s) \triangleq -\frac{1}{N}\sum_{j=1}^{N}\left[\frac{\sum_{i=1}^{d}\left(\lambda_i^{A_j}\right)^4\langle s, U_i^{A_j}\rangle^2}{\sum_{i=1}^{d}\left(\lambda_i^{A_j}\right)^2\langle s, U_i^{A_j}\rangle^2}\right]$$

**The importance of robust solutions**  We start by observing that if $s$ minimizes the training loss $\hat{\mathsf{L}}$, it is not necessarily true that $s$ is the optimal solution for the population loss $\mathsf{L}$. For example, it could be the case that $\{A_j\}_{j=1,\cdots,N}$ are diagonal matrices with only 1 non-zeros on the top row, while $s = (\epsilon, \sqrt{1-\epsilon^2}, 0, \cdots, 0)$ for $\epsilon$ close to 0. In this case, we know that $\hat{\mathsf{L}}_{\mathsf{Tr}}(s) = -1$, which is at its minimum value.

However, such a solution is not robust. In the population distribution, if there exists a matrix $A$ such that $A = \mathrm{diag}(\sqrt{1-100\epsilon^2}, 10\epsilon, 0, 0, \cdots, 0)$, insert $s$ into (3),

$$\frac{\sum_{i=1}^{d}\left(\lambda_i^A\right)^4\langle s, U_i^A\rangle^2}{\sum_{i=1}^{d}\left(\lambda_i^A\right)^2\langle s, U_i^A\rangle^2} = \frac{(1-100\epsilon^2)^2\epsilon^2 + 10^4\epsilon^4(1-\epsilon^2)}{(1-100\epsilon^2)\epsilon^2 + 100\epsilon^2(1-\epsilon^2)} < \frac{\epsilon^2 + 10^4\epsilon^4}{101\epsilon^2 - 100\epsilon^4} = \frac{1 + 10^4\epsilon^2}{101 - 100\epsilon^2}$$

The upper bound is very close to 0 if $\epsilon$ is small enough. This is because when the denominator is extremely small, the whole expression is susceptible to minor perturbations on $A$. This is a typical example showing the importance of finding a *robust* solution. Because of this issue, we will show a generalization guarantee for a *robust* solution $s$.

**Definition of robust solution**  First, define event $\zeta_{A,\delta,s} \triangleq \mathbf{1}\left[\sum_{i=1}^{d}\left(\lambda_i^A\right)^2\langle s, U_i^A\rangle^2 < \delta\right]$, which is the denominator in the loss function. Ideally, we want this event to happen with a small probability, which indicates that for most matrices, the denominator is large, therefore $s$ is robust in general. We have the following definition of robustness.

**Definition 2** (($\rho, \delta$)-robustness). *$s$ is ($\rho, \delta$)-robust with respect to $\mathsf{D}$ if $\mathbf{E}_{A\sim\mathsf{D}}[\zeta_{A,\delta,s}] \leq \rho$. $s$ is ($\rho, \delta$)-robust with respect to $\mathsf{Tr}$ if $\mathbf{E}_{A\sim\mathsf{Tr}}[\zeta_{A,\delta,s}] \leq \rho$.*

For a given $\mathsf{D}$, we can define robust solution set that includes all robust vectors.

**Definition 3** (($\rho, \delta$)-robust set). *$\mathbf{M}_{\mathsf{D},\rho,\delta}$ is defined to be the set of all vectors $s \in \mathbf{S}^{d-1}$ s.t. $s$ is ($\rho, \delta$)-robust with respect to $\mathsf{D}$.*

**Estimating $\mathbf{M}_{\mathsf{D},\rho,\delta}$**  The drawback of the above definition is that $\mathbf{M}_{\mathsf{D},\rho,\delta}$ is defined by the unknown distribution $\mathsf{D}$, so for fixed $\delta$ and $\rho$, we cannot tell whether $s$ is in $\mathbf{M}_{\mathsf{D},\rho,\delta}$ or not. However, we can estimate the robustness of $s$ using the training set. Specifically, we have the following lemma:

**Lemma 4** (Estimating robustness). *For a training set $\mathsf{Tr}$ of size $N$ sampled uniformly at random from $\mathsf{D}$, and a given $s \in \mathbb{R}^d$, a constant $1 > \eta > 0$, if $s$ is ($\rho, \delta$)-robust with respect to $\mathsf{Tr}$, then with probability at least $1 - e^{-\frac{\eta^2\rho N}{2}}$, $s$ is $\left(\frac{\rho}{1-\eta}, \delta\right)$-robust with respect to $\mathsf{D}$.*

*Proof.* Suppose that $\mathrm{Pr}_{A\sim\mathsf{D}}[\zeta_{A,\delta,s}] = \rho_1$, which means $\mathbf{E}\left[\sum_{A_i\in\mathsf{Tr}}\zeta_{A_i,\delta,s}\right] = \rho_1 N$. Since events $\zeta_{A_i,\delta,s}$'s are 0-1 random variables, by Chernoff bound,

$$\mathrm{Pr}\left(\sum_{A_i\in\mathsf{Tr}}\zeta_{A_i,\delta,s} \leq (1-\eta)\rho_1 N\right) \leq e^{-\frac{\eta^2\rho_1 N}{2}}$$

If $\rho_1 < \rho < \rho/(1-\eta)$, our claim is immediately true. Otherwise, we know $e^{-\frac{\eta^2 \rho_1 N}{2}} \leq e^{-\frac{\eta^2 \rho N}{2}}$. Hence, with probability at least $1 - e^{-\frac{\eta^2 \rho N}{2}}$, $N\rho = \sum_{A_i \sim \mathsf{Tr}} \zeta_{A_i, \delta, s} > (1-\eta)\rho_1 N$. This implies that with probability at least $1 - e^{-\frac{\eta^2 \rho N}{2}}$, $\rho_1 \leq \frac{\rho}{1-\eta}$. $\qquad\square$

Lemma 4 implies that for a fixed solution $s$, if it is $(\rho, \delta)$-robust in $\mathsf{Tr}$, it is also $(O(\rho), \delta)$-robust in $\mathsf{D}$ with high probability. However, Lemma 4 only works for a single solution $s$, but there are infinitely many potential $s$ on the $d$-dimensional unit sphere.

To remedy this problem, we discretize the unit sphere to bound the number of potential solutions. Classical results tell us that discretizing the unit sphere into a grid of edge length $\frac{\epsilon}{\sqrt{d}}$ gives $\frac{C}{\epsilon^d}$ points on the grid for some constant $C$ (e.g., see Section 3.3 in [Har-Peled et al., 2012] for more details). We will only consider these points as potential solutions, denoted as $\hat{\mathbf{B}}^d$. Thus, we can find a "robust" solution $s \in \hat{\mathbf{B}}^d$ with decent probability, using Lemma 4 and union bound.

**Lemma 5** (Picking robust $s$). *For a fixed constant $\rho > 0, 1 > \eta > 0$, with probability at least $1 - \frac{C}{\epsilon^d} e^{-\frac{\eta^2 \rho N}{2}}$, any $(\rho, \delta)$-robust $s \in \hat{\mathbf{B}}^d$ with respect to $\mathsf{Tr}$ is $\left(\frac{\rho}{1-\eta}, \delta\right)$-robust with respect to $\mathsf{D}$.*

Since we are working on the discretized solution, we need a new definition of robust set.

**Definition 4** (Discretized $(\rho, \delta)$-robust set). *$\hat{\mathbf{M}}_{\mathsf{D}, \rho, \delta}$ is defined to be the set of all vector $s \in \hat{\mathbf{B}}^d$ s.t. $s$ is $(\rho, \delta)$-robust with respect to $\mathsf{D}$.*

Using similar arguments as Lemma 5, we know all solutions from $\hat{\mathbf{M}}_{\mathsf{D}, \rho, \delta}$ are robust with respect to $\mathsf{Tr}$ as well.

**Lemma 6.** *With probability at least $1 - \frac{C}{\epsilon^d} e^{-\frac{\eta^2 \rho N}{3}}$, for a constant $\eta > 0$, all solutions in $\hat{\mathbf{M}}_{\mathsf{D}, \rho, \delta}$, are $((1+\eta)\rho, \delta)$-robust with respect to $\mathsf{Tr}$.*

*Proof.* Consider a fixed solution $s \in \hat{\mathbf{M}}_{\mathsf{D}, \rho, \delta}$. Note that $\mathbf{E}\left[\sum_{A_i \in \mathsf{Tr}} \zeta_{A_i, \delta, s}\right] = \rho N$ and $\zeta_{A_i, \delta, s}$ are 0-1 random variables. Therefore by Chernoff bound,

$$\Pr\left(\sum_{A_i \in \mathsf{Tr}} \zeta_{A_i, \delta, s} \geq (1+\eta)\rho N\right) \leq e^{-\frac{\eta^2 \rho N}{3}}.$$

Hence, with probability at least $1 - e^{-\frac{\eta^2 \rho N}{3}}$, $s$ is $((1+\eta)\rho, \delta)$-robust with respect to $\mathsf{Tr}$.

By union bound on all points in $\hat{\mathbf{M}}_{\mathsf{D}, \rho, \delta} \subseteq \hat{\mathbf{B}}^d$, the proof is complete. $\qquad\square$

## C.1 Generalization bound

Finally, we show the generalization bounds for robust solutions,. To this can we use Rademacher complexity to prove generalization bound. Define Rademacher complexity $R(\hat{\mathbf{M}}_{\mathsf{D}, \rho, \delta} \circ \mathsf{Tr})$ as

$$R(\hat{\mathbf{M}}_{\mathsf{D}, \rho, \delta} \circ \mathsf{Tr}) \triangleq \frac{1}{N} \mathop{\mathbf{E}}_{\sigma \sim \{\pm 1\}^N} \sup_{s \in \hat{\mathbf{M}}_{\mathsf{D}, \rho, \delta}} \sum_{j=1}^{N} \left[ \frac{\sigma_j \sum_{i=1}^{d} \left(\lambda_i^{A_j}\right)^4 \langle s, U_i^{A_j}\rangle^2}{\sum_{i=1}^{d} \left(\lambda_i^{A_j}\right)^2 \langle s, U_i^{A_j}\rangle^2} \right].$$

$R(\hat{\mathbf{M}}_{\mathsf{D}, \rho, \delta} \circ \mathsf{Tr})$ is handy, because we have the following theorem (notice that the loss function takes value in $[-1, 0]$):

**Theorem 2** (Theorem 26.5 in [Shalev-Shwartz and Ben-David, 2014]). *Given constant $\delta > 0$, with probability of at least $1 - \delta$, for all $s \in \hat{\mathbf{M}}_{\mathsf{D}, \rho, \delta}$,*

$$\mathsf{L}(s) - \hat{\mathsf{L}}_{\mathsf{Tr}}(s) \leq 2R(\hat{\mathbf{M}}_{\mathsf{D}, \rho, \delta} \circ \mathsf{Tr}) + 4\sqrt{\frac{2\log(4/\delta)}{N}}$$

That means, it suffices to bound $R(\hat{\mathbf{M}}_{\mathsf{D}, \rho, \delta} \circ \mathsf{Tr})$ to get the generalization bound. We have the following Lemma.

**Lemma 7** (Bound on $R(\hat{\mathbf{M}}_{\mathsf{D},\rho,\delta} \circ \mathsf{Tr})$). *For a constant $\eta > 0$, with probability at least $1 - \frac{C}{\epsilon^d} e^{-\frac{\eta^2 pN}{3}}$,* $R(\hat{\mathbf{M}}_{\mathsf{D},\rho,\delta} \circ \mathsf{Tr}) \leq (1+\eta)\rho + \frac{1-\delta}{2\delta} + \frac{d}{\sqrt{N}}$.

*Proof.* Define $\rho' = (1+\eta)\rho$. By Lemma 6, we know that with probability $1 - \frac{C}{\epsilon^d} e^{-\frac{\eta^2 pN}{3}}$, any $s \in \hat{\mathbf{M}}_{\mathsf{D},\rho,\delta}$ is $(\rho', \delta)$-robust with respect to $\mathsf{Tr}$, hence $\sum_{A \in \mathsf{Tr}} \zeta_{A,\delta,s} \leq \rho' N$. The analysis below is conditioned on this event.

Define $h_{A,\delta,s} \triangleq \max\{\delta, \sum_{i=1}^{d} (\lambda_i^A)^2 \langle s, U_i^A \rangle^2\}$. We know that with probability $1 - \frac{C}{\epsilon^d} e^{-\frac{\eta^2 pN}{3}}$,

$$N \cdot R(\hat{\mathbf{M}}_{\mathsf{D},\rho,\delta} \circ \mathsf{Tr}) = \mathbf{E}_{\sigma \sim \{\pm 1\}^N} \sup_{s \in \hat{\mathbf{M}}_{\mathsf{D},\rho,\delta}} \sum_{j=1}^{N} \frac{\sigma_j \sum_{i=1}^{d} \left(\lambda_i^{A_j}\right)^4 \langle s, U_i^{A_j} \rangle^2}{\sum_{i=1}^{d} \left(\lambda_i^{A_j}\right)^2 \langle s, U_i^{A_j} \rangle^2}$$

$$\leq \rho' N + \mathbf{E}_{\sigma} \sup_{s \in \hat{\mathbf{M}}_{\mathsf{D},\rho,\delta}} \sum_{j=1}^{N} \frac{\sigma_j \sum_{i=1}^{d} \left(\lambda_i^{A_j}\right)^4 \langle s, U_i^{A_j} \rangle^2}{h_{A,\delta,s}} \tag{5}$$

where (5) holds because by definition, $h_{A,\delta,s} \geq \sum_{i=1}^{d} (\lambda_i^A)^2 \langle s, U_i^A \rangle^2$ if and only if $\zeta_{A,\delta,s} = 1$, which happens for at most $\rho' N$ matrices. Note that for any matrix $A_j$, $\frac{\sigma_j \sum_{i=1}^{d} \left(\lambda_i^{A_j}\right)^4 \langle s, U_i^{A_j} \rangle^2}{\sum_{i=1}^{d} \left(\lambda_i^{A_j}\right)^2 \langle s, U_i^{A_j} \rangle^2} \leq 1$.

Now,

$$\mathbf{E}_{\sigma} \sup_{s \in \hat{\mathbf{M}}_{\mathsf{D},\rho,\delta}} \sum_{j=1}^{N} \frac{\sigma_j \sum_{i=1}^{d} \left(\lambda_i^{A_j}\right)^4 \langle s, U_i^{A_j} \rangle^2}{h_{A,\delta,s}}$$

$$\leq \mathbf{E}_{\sigma} \sup_{s \in \hat{\mathbf{M}}_{\mathsf{D},\rho,\delta}} \sum_{j=1}^{N} \left( \mathbf{1}_{\sigma_j = 1} \frac{\sum_{i=1}^{d} \left(\lambda_i^{A_j}\right)^4 \langle s, U_i^{A_j} \rangle^2}{\delta} - \mathbf{1}_{\sigma_j = -1} \sum_{i=1}^{d} \left(\lambda_i^{A_j}\right)^4 \langle s, U_i^{A_j} \rangle^2 \right) \tag{6}$$

$$= \mathbf{E}_{\sigma} \sup_{s \in \hat{\mathbf{M}}_{\mathsf{D},\rho,\delta}} \sum_{j=1}^{N} \left( \sigma_j \sum_{i=1}^{d} \left(\lambda_i^{A_j}\right)^4 \langle s, U_i^{A_j} \rangle^2 + \mathbf{1}_{\sigma_j = 1} \sum_{i=1}^{d} \left(\lambda_i^{A_j}\right)^4 \langle s, U_i^{A_j} \rangle^2 \left(\frac{1}{\delta} - 1\right) \right)$$

$$\leq \frac{N}{2\delta} - \frac{N}{2} + \mathbf{E}_{\sigma} \sup_{s \in \hat{\mathbf{M}}_{\mathsf{D},\rho,\delta}} \sum_{j=1}^{N} \sigma_j \sum_{i=1}^{d} \left(\lambda_i^{A_j}\right)^4 \langle s, U_i^{A_j} \rangle^2 \tag{7}$$

The first inequality, (6), holds as $\frac{\sum_{i=1}^{d} \left(\lambda_i^{A_j}\right)^4 \langle s, U_i^{A_j} \rangle^2}{h_{A,\delta,s}} \in [\delta, 1]$. It remains to bound the last term (7).

$$\mathbf{E}_{\sigma} \sup_{s \in \hat{\mathbf{M}}_{\mathsf{D},\rho,\delta}} \sum_{j=1}^{N} \sigma_j \sum_{i=1}^{d} \left(\lambda_i^{A_j}\right)^4 \langle s, U_i^{A_j} \rangle^2 \leq \sum_{i=1}^{d} \mathbf{E}_{\sigma} \sup_{s \in \hat{\mathbf{M}}_{\mathsf{D},\rho,\delta}} \sum_{j=1}^{N} \sigma_j \left\langle s, \left(\lambda_i^{A_j}\right)^2 U_i^{A_j} \right\rangle^2 \tag{8}$$

By contraction lemma of Rademacher complexity, we have

$$\mathbf{E}_{\sigma} \sup_{s \in \hat{\mathbf{M}}_{\mathsf{D},\rho,\delta}} \sum_{j=1}^{N} \sigma_j \left\langle s, \left(\lambda_i^{A_j}\right)^2 U_i^{A_j} \right\rangle^2 \leq \mathbf{E}_{\sigma} \sup_{s \in \hat{\mathbf{M}}_{\mathsf{D},\rho,\delta}} \sum_{j=1}^{N} \sigma_j \left\langle s, \left(\lambda_i^{A_j}\right)^2 U_i^{A_j} \right\rangle$$

$$= \mathbf{E}_{\sigma} \sup_{s \in \hat{\mathbf{M}}_{\mathsf{D},\rho,\delta}} \left\langle s, \sum_{j=1}^{N} \sigma_j \left(\lambda_i^{A_j}\right)^2 U_i^{A_j} \right\rangle$$

$$\leq \mathbf{E}_{\sigma} \left\| \sum_{j=1}^{N} \sigma_j \left(\lambda_i^{A_j}\right)^2 U_i^{A_j} \right\|_2$$

Where the last inequality is by Cauchy-Schwartz inequality. Now, using Jensen's inequality, we have

$$\mathbf{E}_\sigma \left\| \sum_{j=1}^N \sigma_j \left( \lambda_i^{A_j} \right)^2 U_i^{A_j} \right\|_2 \leq \left( \mathbf{E}_\sigma \left\| \sum_{j=1}^N \sigma_j \left( \lambda_i^{A_j} \right)^2 U_i^{A_j} \right\|_2^2 \right)^{1/2} = \left( \sum_{j=1}^N \left( \lambda_i^{A_j} \right)^4 \right)^{1/2} \leq \sqrt{N} \tag{9}$$

Combining (5), (7), (8) and (9), we have $R(\hat{\mathbf{M}}_{\mathsf{D},\rho,\delta} \circ \mathsf{Tr}) \leq \rho' + \frac{1-\delta}{2\delta} + \frac{d}{\sqrt{N}}$. $\qquad\square$

Combining with Theorem 2, we get our main theorem:

**Theorem 3** (Main Theorem). *Given a training set* $\mathsf{Tr} = \{A_j\}_{j=1}^N$ *sampled uniformly from* $\mathsf{D}$, *and fixed constants* $1 > \rho \geq 0, \delta > 0, 1 > \eta > 0$, *if there exists a* $(\rho, \delta)$-*robust solution* $s \in \hat{\mathbf{B}}^d$ *with respect to* $\mathsf{Tr}$, *then with probability at least* $1 - \frac{C}{\epsilon^d} e^{-\frac{\eta^2 \rho N}{2}} - \frac{C}{\epsilon^d} e^{-\frac{\eta^2 \rho N}{3(1-\eta)}}$, *for* $s \in \hat{\mathbf{B}}^d$ *that is a* $(\rho, \delta)$-*robust solution with respect to* $\mathsf{Tr}$,

$$\mathsf{L}(s) \leq \hat{\mathsf{L}}_{\mathsf{Tr}}(s) + \frac{2(1+\eta)\rho}{1-\eta} + \frac{1-\delta}{\delta} + \frac{2d}{\sqrt{N}} + 4\sqrt{\frac{2\log(4/\delta)}{N}}$$

*Proof.* Since we can find $s \in \hat{\mathbf{B}}^d$ s.t. $s$ is $(\rho, \delta)$-robust with respect to $\mathsf{Tr}$, by Lemma 5, with probability $1 - \frac{C}{\epsilon^d} e^{-\frac{\eta^2 \rho N}{2}}$, $s$ is $(\frac{\rho}{1-\eta}, \delta)$-robust with respect to $\mathsf{D}$. Therefore, $s \in \hat{\mathbf{M}}_{\mathsf{D}, \frac{\rho}{1-\eta}, \delta}$. Apply Lemma 7, we have With probability at least $1 - \frac{C}{\epsilon^d} e^{-\frac{\eta^2 \rho N}{3(1-\eta)}}$, $R(\hat{\mathbf{M}}_{\mathsf{D},\rho,\delta} \circ \mathsf{Tr}) \leq \frac{\rho(1+\eta)}{1-\eta} + \frac{1-\delta}{2\delta} + \frac{d}{\sqrt{N}}$. Combined with Theorem 2, the proof is complete. $\qquad\square$

In summary, Theorem 3 states that if we can find a solution $s$ which "fits" the training set, and is very robust, then it generalizes to the test set.

## D Missing Proofs of Section 4

**Fact 1** (Pythagorean Theorem). *If $A$ and $B$ are matrices with the same number of rows and columns, then $AB^\top = 0$ implies $||A + B||_F^2 = ||A||_F^2 + ||B||_F^2$.*

*Proof of Lemma 1.* Note that $AVV^\top$ is a row projection of $A$ on the $\mathrm{colsp}(V)$. Then, for any conforming $Y$,

$$(A - AVV^\top)(AVV^\top - YV^\top)^\top = A(I - VV^\top)V(AV - Y)^\top$$
$$= A(V - VV^\top V)(AV - Y)^\top = 0.$$

where the last equality follows from the fact if $V$ has orthonormal columns then $VV^\top V = V$ (e.g., see Lemma 3.5 in [Clarkson and Woodruff, 2009]). Then, by the Pythagorean Theorem (Fact 1), we have

$$||A - YV^\top||_F^2 = ||A - AVV^\top||_F^2 + ||AVV^\top - YV^\top||_F^2 \tag{10}$$

Since $V$ has orthonormal columns, for any conforming $x$, $||x^\top V^\top|| = ||x||$. Thus, for any $Z$ of rank at most $k$,

$$||AVV^\top - [AV]_k V^\top||_F = ||(AV - [AV]_k)V^\top||_F = ||AV - [AV]_k||_F$$
$$\leq ||AV - Z||_F = ||AVV^\top - ZV^\top||_F \tag{11}$$

Hence,

$$||A - [AV]_k V^\top||_F^2 = ||A - AVV^\top||_F^2 + ||AVV^\top - [AV]_k V^\top||_F^2 \quad \triangleright \text{By (10)}$$
$$\leq ||A - AVV^\top||_F^2 + ||AVV^\top - ZV^\top||_F^2 \quad \triangleright \text{By (11)}$$

This implies that $[AV]_k V^\top$ is a best rank-$k$ approximation of $A$ in the $\mathrm{colsp}(V)$. $\qquad\square$