[Reviews · NeurIPS 2019]

Reviewer 1



After Response: Overall, I found the authors' response to provide sufficient clarification. Most of my concerns were over the presentation of the material, and I am satisfied with their clarifications and commitment to polishing the paper a bit. I am confident in the correctness of the work and believe it is of general interest. ----------------------- Original Comments: Originality: I don't know the literature on sketching based low rank approximation to comment on the novelty of this work, but relevant literature appears adequately cited so I have no reason to suspect it is not new. Quality: The submission is technically sound with a bit of theory and some good experimental results. I have made suggestions for expanding the work a bit. Clarity: The paper adequately informs the reader. I have made comments below about reorganizing the paper a bit. Significance: The method appears new and concerns two problems of interest, namely low rank approximation and sketching. The empirical results are compelling. It would likely be of interest to researchers working in these areas. The authors propose a method for using learned sketching matrices to find low rank approximations. The experimental results look good and they provide a bound for worst case performance. Experimental comments: I think it is necessary to include the number of runs you are using for these experiments, add error bars to the plots in figures 2 and 3, and to report the standard error for tables 1 and 2. This could potentially illustrate another possible advantage of your method, as sketching methods tend to increase variance (of the estimators) in favor of computational complexity and it seems possible that using a fully or partially learned sketching matrix could decrease variance. Also, it would be nice if you bolded the best performance for each test in tables 1 and 2. Since part of the motivation for sketching is to reduce computational complexity, you should report run-times in some form or add a dedicated experiment to showcase the run-time of your approach. At minimum, I think you should spell out the computational complexity of unsketched low rank approximation, sparse/dense random sketched low rank approximation, as well as your method for fully and partially learned sketching matrices. Technical comments: Regarding my variance comment from the experimental comments section, it could be possible (or maybe even straightforward?) to prove this. For one paper that discusses issues of variance and bias in sketching methods see: “Sketched Ridge Regression…” (Wang et al. 2018) Since computational efficiency for low rank approximations is a main motivation for this paper, I think it is necessary to state the computational complexity of the SCW algorithm and the overall complexity of the proposed approach as well. Line 152: Even if you are not computing gradients analytically, I think it is very important to give a very explicit description of your end-to-end process. This is probably most effectively done through some sort of block diagram type graphic. Lines 154: Autograd is distinct from numerical differentiation. Autograd essentially invokes chain rule to compute the exact derivatives up to machine precision. Also, when python has a dedicated autograd library it seems like overkill to use PyTorch, but this probably isn’t so important for your paper. Lines 156-158: I am confused by this paragraph. In lines 125-126 you imply that Algorithm 1 IS SCW, then here you are saying they are different. Do you mean to say Algorithm 2 in line 157? Also, If the running time is the same, you should specify in what way Algorithm 1 has better performance. Section 4: You should specify this is a worst case bound for the “mixed” S. Overall I really like the idea. It is a simple and creative solution to a possible big issue of using a learned sketching matrix in this context. Also, I think you should explicitly state exactly what this result implies somewhere in the beginning of this section. Line 184: What is Z? Do you mean V_1? Writing comments: In general the writing is grammatically and stylistically pretty good. There are a few spots that could use another proof reading. For example: Lines 149-151: The last two sentences of this paragraph could be improved a bit in terms of clarity and brevity. Also, I think the paper could be reorganized a bit for greater clarity, especially since there is some wiggle room to fill the 8 page limit. Specifically, it would be nice if you spelled out exactly what your method is doing in some sort of block diagram. I don’t think it is necessary to write out the gradients or anything, but I do think you should write out exactly what loss you are minimizing over what variable, what set that variable lives in, and whether or not you use the same training set for both learning S and also using that S to calculate the best rank k approximation.

Reviewer 2



Updated Comments Following Authors' Response : I thank the authors for their responses and extending their experiments according to the comments. For the experiment on the effect of the number of rows on test error, I think it would be interesting to view some plots to see if there's an elbow, which might indicate a good trade-off between computational savings and reconstruction error. I had noted that I'd be willing to increase my score if the authors extend their experiments, and I did. ******* The authors consider the problem of low rank approximation of a matrix A, using a 'sketching-matrix' based method. The idea is to multiply A with a random matrix S, so as to reduce the number of rows of A, and obtain an approximation using the SVD of SA, instead of A. The authors argue that a 'learned' S can do potentially better than a random one. They also note that a learned S may not have a good worst-case performance, and they consider a hybrid approach where a portion of the matrix (some of the rows) is randomly selected and fixed during learning (learning may or may not have information about the fixed rows, giving two different cases). They demonstrate their claims experimentally. The learning is performed using stochastic gradient descent, where the gradients are computed numerically by PyTorch. In order to obtain a differentiable 'SVD' operation, that appears in the expression for the cost function, the authors resort to a method based on power iterations (with a fixed number of iterations, T). In terms of theoretical analysis, the authors note that by adding rows to S, one obtains a better approximation. Apart from this modest result, they also provide in the appendix some worst case analysis for the case where S has a single row. However, even though this analysis is mentioned in the introduction, this issue is not treated at all in the rest of the main text. Overall, the idea looks potentially interesting, but I think the experiments can be modified to better demonstrate how useful this scheme can be in practice. Here are some specific comments : - In the statement of Thm 1, should V_* and V_1 be transposed? - Doesn't Claim 1 say that if you increase the search space in an optimization problem, you get a better solution? I don't think you need to include that as a separate claim. - For the video experiments, do you obtain a different S for each dataset? That is, is the S used for Logo different than the one for Friends, etc.? I got the impression that this is the case, and the following comments are based on this understanding. In fact, I was expecting to see an experiment where training is performed on a vast collection of images, and a single 'S' matrix to be proposed for that collection. With the current setup, isn't the utility of the idea limited? What is the typical application you have in mind? - If we were to use the three video sets (Logo, Friends, Eagle) to obtain a single sketching matrix S, does that S still lead to a much better approximation than that from a random S? - The worst case performance is addressed in the experiments. I think one way to obtain an idea might be to use an S computed for Logo, say, and use it to obtain an approximation from an image from Eagle, for instance. Then you could compare that with random/mixed etc. - Is there any structure on the learned skecthing matrix S? If so, is that structure more or less common across different video sets? - The number of random vs learned rows appears to be the same in the experiments, and this still gives decent performance. If we reduce the ratio of the number of learned rows, how does the error curve with respect to the ratio change? If one could get away with a good performance when only a single row is learned, that would save computation time for learning (even though that's done offline).

Reviewer 3



This paper studies an interesting problem of using machine learned advice to speed up low rank approximation. Rigorous theoretical analysis and sound experimental results over proposed method are provided. This work is a generalized work of ‘Learning-Based Frequency Estimation Algorithms’. Besides, low rank approximation is generally mixed with other problems, such as RPCA, where iteration optimization method is used to solve the involved model. Hence, the proposed learning based method, an iteration method, may limits the efficiency of entire optimization procedure. The author should compare the time computations of SVD、proposed method、sketch method.

[Author Response · NeurIPS 2019]

We thank all the reviewers for helpful comments and insights. We will incorporate suggestions about writing, diagrams and presentation in the final version of the paper. In what follows we include experiments requested by the reviewers, albeit on a small scale (80 matrices for training and 20 for testing), so the numbers might be slightly different from those in the paper. We will include full scale experiments in the final version of the paper.

**To Reviewer 1**:

- **Error bars and standard errors.** Currently the error is averaged over multiple test matrices (as described in lines 208-218), but the matrix is learned/generated only once. For this rebuttal we computed the matrix 5 times, on Logo, with $k = 10, m = 20$ and 100 matrices. The variance of our learned algorithm is indeed lower.

|  | Learned | Random |
|---|---|---|
| Avg Test Error | 0.0629 | 2.264 |
| Standard Error | 0.0013 | 0.163 |

- **Complexity of the prior/proposed algorithms.** Our main contribution is a learning-augmented version of the streaming algorithm due to Sarlos/Clarkson-Woodruff (SCW). Since we augment a streaming algorithm, our main focus is on improving its *space* usage (which in the distributed setting translates into the amount of communication). The latter is $O(md)$, the size of $SA$. Since we optimize the tradeoff between $m$ and the accuracy, we directly improve the space usage and/or accuracy of the algorithm.

  Regarding the *time* complexity of SCW algorithm, it is $O(nmd)$ assuming $k \le m \le \min(n, d)$.

- **Line 156-158.** SCW proposed Alg 1 with random sketching matrices S. In this paper, we follow the same framework but instead we use learned (or partially learned) matrices. Our empirical results show that the **performance**–the quality of the returned rank-$k$ matrix–of our algorithm (i.e., Alg 1 with learned matrices) is better than of the SCW algorithm when the training distribution is close to the test distribution.

**To Reviewer 3**:

- **Use three video sets to learn $S$.** Yes, right now we learn different $S$ for each data set. This reflects our intended applications such as processing satellite imaging data or video monitoring frames, where the data distribution does not change much between uses.

  We have run a small experiment to learn a *single* matrix $S$ with $m = k = 10$ (300 matrices) on Logo, Eagle and Friends (100 matrices each). We then tested it on Logo and compared to a learned $S$ using only using 100 Logo matrices, as well as to a random $S$. The mixed case is close to Logo and much better than Random.

|  | Logo+Eagle+Friends | Logo only | Random |
|---|---|---|---|
| Test Error | 0.67 | 0.27 | 5.19 |

- **Vary the ratio of random vs learned rows.** Alas, there is no free lunch. If the number of learned rows decreases, the test error will increase. We have run a small experiment, with $m = 20, k = 10$ using 100 Logo matrices, running for 3000 iterations. See the results below. The error curve seems monotone, and there exists a tradeoff between #learned rows and the error.

| #Learned Rows | 1 | 2 | 3 | 5 | 8 | 12 | 15 | 17 | 18 | 19 |
|---|---|---|---|---|---|---|---|---|---|---|
| Test Error | 1.43 | 1.18 | 1.06 | 0.783 | 0.281 | 0.119 | 0.090 | 0.081 | 0.064 | 0.068 |

- **Worst case construction.** As suggested, we tried to use S computed for Logo, and test it on Eagle. In this experiment, we use $m = k = 10$ and 100 matrices. The sketch matrix learned using Logo works worse on Eagle, as expected, but still works better than Random. This could be because both Logo and Eagle are video datasets.

|  | Logo-learned | Eagle-learned | Random |
|---|---|---|---|
| Test Error | 0.68 | 0.16 | 7.38 |

- **Common structure on $S$.** This is a good question! We have investigated it, but haven't got clear answers so far.

**To Reviewer 5**:

- **Running time.** We report them for Logo matrices with $m = k = 10$. It can be seen that sketching methods are much faster than the "standard" SVD. Notice that training only needs to be done *once*, and can be done offline.

| SVD | SCW | Our-Inference | Our-Training |
|---|---|---|---|
| 2.2s | 0.03s | 0.03s | 9481.25s |

- **Experiment over RPCA.** Thank you for your suggestion! Unfortunately, we did not get enough time to run the RPCA experiment.

[Meta-Review · NeurIPS 2019]

Learning sketches for low-rank approximation is a problem that many people have been interested in. This paper makes a nice original contribution to the problem. The worst-case bound is interesting, and the empirical results are compelling. I encourage the authors to address reviewer comments in the paper.